Impact of preoperative underweight status on perioperative morbidity and mortality after liver resection for liver tumors: a meta-analysis

Tang Xuan 1 2
Su Yunpeng 1 2
Li Wenxi 1
Wan Li wanli21078@163.com 3
1 Department of Surgery, Mianyang Central Hospital , Mianyang , China
2 Department of Nursing, Dali University , Dali , China
3 Department of Nursing, Mianyang Central Hospital , Mianyang , China
Albuquerque Ulysses
Electronic publication date: 2025 Nov 13
Publication date: 2025
Volume: 13
Electronic Location ID: e20324
Received 2024 Dec 6; Accepted 2025 Oct 10
Copyright: ©2025 Tang et al.
Copyright year: 2025
Copyright holder: Tang et al.
License: This is an open access article distributed under the terms of the Creative Commons Attribution License, which permits unrestricted use, distribution, reproduction and adaptation in any medium and for any purpose provided that it is properly attributed. For attribution, the original author(s), title, publication source (PeerJ) and either DOI or URL of the article must be cited.
License URL: https://creativecommons.org/licenses/by/4.0/

Keywords: Liver tumors, Underweight, Mortality, Morbidity, Meta-analysis

Funding: The authors received no funding for this work.

==============================
Background

The impact of preoperative underweight status on the outcomes of liver resections for liver tumors is debated. We aimed at conducting a meta-analysis to evaluate the associations between underweight and short-term clinical outcomes after liver resection.

Methods

PubMed, Cochrane Library, Embase, Scopus , and Web of Science databases were searched from inception to September 14, 2024 (updated on April 21, 2025) to identify eligible studies. Risk ratios (RRs) or mean differences (MDs) and 95% confidence intervals (CI) were calculated.

Results

Seven studies were included, with a total of 3,835 patients (normal weight group: 3,412 patients; underweight group: 423 patients). Findings indicated that, relative to individuals of normal weight, underweight patients have longer operation time (MD, 7.73 mins; 95% CI [2.08–13.38], P = 0.007) and higher overall postoperative morbidity (RR 1.41 95% CI [1.06–1.88], P = 0.02), mortality (RR 2.98, 95% CI [1.4 3–6.20], P = 0.003), and surgical site infection (RR 2.00, 95% CI [1.03–3.88], P = 0.04). There were no significant differences in blood loss, length of stay, liver failure, bile leak, and blood transfusion between the normal weight and underweight groups.

Conclusions

Preoperative underweight status is associated with higher risk of morbidity and mortality after liver resection. Prospective studies or interventional trials, such as nutritional prehabilitation or stratification by cirrhosis and resection extent, are needed to confirm whether underweight represents a truly modifiable risk factor. This review is intended for professionals within the fields of hepatobiliary surgery.

Introduction

Liver tumors include benign tumors and malignant tumors, and the malignant tumors are mainly hepatocellular carcinoma (HCC) and liver metastases (Zhou et al., 2010). HCC is the sixth most common cancer worldwide, causing nearly 800,000 deaths each year (Bray et al., 2024). Surgical resection is the main treatment strategy for liver tumors (Liu et al., 2022). However, the incidence of complications after liver resection is as high as 20% to 34% (Yoo et al., 2024; Yu et al., 2020; Zhang et al., 2024). Postoperative complications not only prolong the hospitalization time and increase the cost, but also affect the long-term prognosis of patients (Yang et al., 2024). Therefore, identifying the risk factors of postoperative complications to reduce the incidence of postoperative complications is essential to improve patient outcomes.

Previous studies have shown that preoperative body mass index (BMI), especially being overweight or obese, affects postoperative morbidity after liver resection (Liu et al., 2022). A meta-analysis of 14 studies included by Rong et al. (2015) showed that higher BMI was associated with an increased risk of wound infection (risk ratio (RR) = 2.17, 95% confidence interval (CI) [1.28–3.68], P = 0.004). A recent multicenter cohort study (Zimmitti et al., 2022) also confirmed that obese patients had longer operation times, more intraoperative blood loss, and a higher incidence of postoperative complications than non-obese patients. However, the relationship between underweight and the short-term outcome of liver resection remains controversial. As the tumor progresses and nutrients are consumed, underweight status is common in patients with liver tumors (Yang et al., 2024). A retrospective study by Yu et al. (2020) found that underweight status significantly increased the risk of postoperative complications. However, the study by Okamura et al. (2012) showed no significant difference in postoperative morbidity between the low BMI group and the normal BMI group. In addition, there is still a lack of relevant systematic reviews or meta-analyses on this research topic.

Therefore, the aim of this study was to conduct a systematic review of the literature and employ a meta-analysis to assess the effects of underweight on postoperative morbidity and mortality in patients with liver tumors.

Materials & Methods

Search strategy

This study was conducted in accordance with the Preferred Reporting Items for Systematic Reviews and Meta-Analyses (PRISMA) (Page et al., 2021). The study was registered in the PROSPERO database (CRD42024607190).

Two investigators (Xuan Tang and Yunpeng Su) independently conducted a comprehensive literature search using the Web of Science, PubMed, EMBASE, Scopus, and Cochrane Library databases to identify studies published before September 14, 2024 (updated on April 21, 2025). The detailed search strategy is presented in Table 1. In addition, we checked the reference lists of the eligible articles to identify other potential studies. No language restrictions were applied during the search process.

Table 1 Electronic search strategy.

Database	Search term (published up to April 21, 2025)	Number	
PubMed	(Thinness[Mesh] OR Body Mass Index[Mesh] OR low body mass index[Title/Abstract] OR low BMI[Title/Abstract] OR underweight[Title/Abstract] OR low body weight[Title/Abstract]) AND (Hepatectomy[Mesh] OR Hepatectomy[Title/Abstract] OR liver resection[Title/Abstract] OR hepatic resection[Title/Abstract])	155	
Embase	(Thinness OR low body mass index OR low BMI OR underweight OR low body weight).ab,kw,ti. AND (Hepatectomy OR liver resection OR hepatic resection).ab,kw,ti.	48	
Cochrane Library Trials	((Thinness OR low body mass index OR low BMI OR underweight OR low body weight):ti,ab,kw) AND ((Hepatectomy OR liver resection OR hepatic resection):ti,ab,kw)	60	
Scopus	(Thinness OR low body mass index OR low BMI OR underweight OR low body weight).ab,kw,ti. AND (Hepatectomy OR liver resection OR hepatic resection).ab,kw,ti.	122	
Web of Science	(TS=(Thinness OR low body mass index OR low BMI OR underweight OR low body weight)) AND (TS=(Hepatectomy OR liver resection OR hepatic resection))	885	

Study selection

Studies included in this meta-analysis were chosen according to the Patient, Intervention, Comparison, Outcomes, Study Type (PICOS) criteria, as shown below: (1) patient: patients undergoing liver resection for liver tumors. Patients were categorized as normal weight (BMI = 18.5–24.9 kg/m2) and underweight (BMI < 18.5 kg/m2) according to the World Health Organization (WHO) definition (Yang et al., 2024); (2) intervention: patients who are underweight; (3) comparison: patients of normal weight; (4) outcomes: primary outcomes encompassed mortality and overall complications. Secondary outcomes included major complication, operation time, blood loss, surgical site infection (SSI), blood transfusion, liver failure, bile leak, and length of stay. Major complications were defined as Clavien–Dindo grade ≥III. Post-hepatectomy liver failure and bile leak were defined according to International Study Group for Liver Surgery (ISGLS) criteria. Mortality was reported at 30 days and at 90 days; (5) study type: cohort studies and case-control studies.

The exclusion criteria were as follows: case reports, letters, unpublished manuscripts, conference abstracts, and animal studies.

Data extraction

Two authors (Xuan Tang and Yunpeng Su) independently extracted data, which included author name, year of publication, study design, country, study population (sample size and diagnosis), type of liver resection and outcome information (mortality, morbidity, blood loss, operation time, blood transfusion, and length of stay). To avoid overlapping patient cohorts, we cross-checked study centers, recruitment years, and sample sizes. When potential overlap was suspected, the study with the larger cohort or longer follow-up was retained. When data of interest were unavailable, the corresponding author was contacted to obtain the missing data.

Quality assessment

The quality assessment was conducted independently by two authors (Xuan Tang and Yunpeng Su) using the Newcastle-Ottawa Scale (NOS), which assigns a score on a 9-point scale. A score of ≥7 indicates high quality, and scores of 5–6 indicate moderate quality. Any discrepancies were resolved by a third author (Li Wan).

Statistical analysis

The meta-analysis was performed using the Review Manager software (version 5.3). Risk ratios (RRs) with corresponding 95% confidence intervals (CI) were calculated for categorical outcome variables and mean difference (MD) with corresponding 95% CI for continuous outcome variables. Heterogeneity across studies was assessed by using the I2 statistic. All analyses were conducted using the random effects model. To explore the robustness of the primary outcomes, we adopted the one-study exclusion method to evaluate the impact of each study on the total effect size. Publication bias was assessed using Egger’s test and funnel plot (for primary outcomes). Trial sequential analysis (TSA) was conducted using TSA v0.9.5.10. The execution of TSA entailed the establishment of O’ Brien-Fleming a-spending boundaries, which were determined using a 5% type I error and an 80% power. Statistical significance was established at P-value of less than 0.05.

Results

Literature retrieval

Our comprehensive literature search found 1,271 records, of which 249 were duplicates. After reviewing titles and abstracts, 996 papers were excluded, and the full texts of the remaining 26 studies were evaluated. Finally, seven studies (Ishihara et al., 2022; Liu et al., 2022; Okamura et al., 2012; Wang et al., 2014; Yang et al., 2024; Yu et al., 2020; Zhao et al., 2022) were included in the final analysis (Fig. 1).

Figure 1 The PRISMA flowchart.

Study characteristics and quality assessment

The detailed characteristics of the seven studies (Ishihara et al., 2022; Liu et al., 2022; Okamura et al., 2012; Wang et al., 2014; Yang et al., 2024; Yu et al., 2020; Zhao et al., 2022) were described in Table 2. The studies were published between 2012 and 2024 and included 3,835 patients (normal weight group: 3,412 patients; underweight group: 423 patients). Indications for operative management were for malignancy in seven studies (Ishihara et al., 2022; Liu et al., 2022; Okamura et al., 2012; Wang et al., 2014; Yang et al., 2024; Yu et al., 2020; Zhao et al., 2022). The included patients were mainly from China, and Japan. All studies were considered of moderate to high quality, achieving a score of ≥6 based on the NOS (Table 2).

Table 2 Study characteristics of the seven included studies.

First author, year	Design	Period of study	Type of hepatectomy	Sample size	Patients in each BMI group	Diagnosis	NOS	
					Normal weight	Underweight			
Okamura et al. (2012)	RCS	2002–2007	Minor and major hepatectomies	147	125	22	HCC	7/9	
Wang et al. (2014)	RCS	2009–2013	Minor and major hepatectomies	1,058	982	76	HCC	7/9	
Yu et al. (2020)	RCS	2010–2016	Minor and major hepatectomies	841	733	108	HCC	8/9	
Ishihara et al. (2022)	RCS	2000–2019	Minor and major hepatectomies	768	694	74	HCC	7/9	
Liu et al. (2022)	RCS	2015–2018	Minor and major hepatectomies	760	660	100	HCC	8/9	
Zhao et al. (2022)	RCS	2003–2016	Minor and major hepatectomies	119	96	23	HCC	7/9	
Yang et al. (2024)	RCS	2013–2019	Minor and major hepatectomies	142	122	20	HCC	8/9	
Notes.

HCC, hepatocellular carcinoma; RCS, retrospective cohort study.

Meta-analysis

Mortality

Six studies (Ishihara et al., 2022; Liu et al., 2022; Okamura et al., 2012; Wang et al., 2014; Yang et al., 2024; Yu et al., 2020) reported data on mortality. The combined results of the six studies showed that mortality was significantly higher in the underweight group than in the normal weight group (RR 2.98, 95% CI [1.43–6.20]; Heterogeneity: I2 = 0%, P = 0.82) (Fig. 2A). Underweight was associated with an absolute increase of 1% in postoperative mortality, corresponding to a number needed to harm (NNH) of 100. In addition, the 90-day mortality of the underweight group was significantly higher (RR 2.79, 95% CI [1.11–7.03]; Heterogeneity: I2 = 0%, P = 0.66) (Fig. 2B) than that of the normal weight group (Table 3). Underweight was associated with an absolute increase of 2% in 90-day mortality, corresponding to a NNH of 50.

Figure 2 Comparison of primary outcomes between the two groups.

(A) mortality. (B) 90-day mortality. (C) overall morbidity. Note. Ishihara et al., 2022; Liu et al., 2022; Okamura et al., 2012; Wang et al., 2014; Yang et al., 2024; Yu et al., 2020; Zhao et al., 2022.

Morbidity

Seven studies (Ishihara et al., 2022; Liu et al., 2022; Okamura et al., 2012; Wang et al., 2014; Yang et al., 2024; Yu et al., 2020; Zhao et al., 2022) assessed the total postoperative complications. The pooled results showed that being underweight was associated with an increased risk of total postoperative complications (RR 1.41, 95% CI [1.06–1.88]; Heterogeneity: I2 = 68%, P = 0.005) (Fig. 2C). Underweight was associated with an absolute increase of 10% in postoperative morbidity, corresponding to a NNH of 10.

Major complications

Data from five studies (Ishihara et al., 2022; Liu et al., 2022; Wang et al., 2014; Yang et al., 2024; Yu et al., 2020; Zhao et al., 2022) of 2,928 patients did not reveal any difference between the underweight and normal weight groups (RR 1.58, 95% CI [0.87–2.84]; heterogeneity: I2 = 52%, P = 0.08) (Fig. 3A) in terms of major complications.

Blood loss

Four studies (Ishihara et al., 2022; Wang et al., 2014; Yang et al., 2024; Zhao et al., 2022) provided information on intraoperative blood loss. The combined results showed that the underweight group (Fig. 4D) had similar intraoperative blood loss as the normal weight group (MD, 32.43 mL; 95% CI [−80.65, 145.50], P = 0.57; heterogeneity: I2 = 85%) (Fig. 3B).

Operation time

The operation time was reported in four trials (Ishihara et al., 2022; Okamura et al., 2012; Yang et al., 2024; Zhao et al., 2022). The combined results showed that being underweight (MD, 7.73 mins; 95% CI [2.08–13.38], P = 0.007) (Fig. 3C) was associated with prolonged operation time.

Length of stay

The length of the hospital stay was reported in seven studies (Ishihara et al., 2022; Liu et al., 2022; Okamura et al., 2012; Wang et al., 2014; Yang et al., 2024; Yu et al., 2020; Zhao et al., 2022). According to the results of this meta-analysis, there was no significant difference in length of hospital stay between the underweight group and the normal weight group (MD, 0.47 days; 95% CI [−0.01, 0.96], P = 0.06) (Fig. 3D).

Table 3 Summary of results from all outcomes.

Outcomes	No. of studies	Events for underweight group	Events for normal weight group	Effect size	95% CI	P	I2 (%)	
Overall complications	7	140/423	888/3,412	1.41	1.06, 1.88	0.02	68	
Mortality	6	10/400	29/3,316	2.98	1.43, 6.20	0.003	0	
90-day Mortality	2	6/176	20/1,642	2.79	1.11, 7.03	0.03	0	
Major complications	5	32/301	218/2,627	1.58	0.87, 2.84	0.13	52	
Surgical site infection	4	28/224	120/1,674	2.00	1.03, 3.88	0.04	51	
Bile leak	6	19/323	114/2,752	1.35	0.84, 2.19	0.22	0	
Liver failure	4	14/278	133/2,531	1.12	0.65, 1.93	0.68	0	
Blood transfusion	4	57/218	409/1,889	1.14	0.74, 1.77	0.55	64	
Blood loss	4	–	–	32.43	−80.65, 145.50	0.57	85	
Operation time	4	–	–	7.73	2.08, 13.38	0.007	0	
Hospital stay	7	–	–	0.47	−0.01, 0.96	0.06	0	

Figure 3 Comparison of secondary outcomes between the two groups.

(A) major complications (B) intraoperation blood loss. (C) operation time. (D) length of stay. Note. Ishihara et al., 2022; Liu et al., 2022; Okamura et al., 2012; Wang et al., 2014; Yang et al., 2024; Yu et al., 2020; Zhao et al., 2022.

Figure 4 Comparison of secondary outcomes between the two groups.

(A) liver failure. (B) bile leak. (C) surgical site infection. (D) blood transfusion. Note. Ishihara et al., 2022; Liu et al., 2022; Okamura et al., 2012; Wang et al., 2014; Yang et al., 2024; Yu et al., 2020; Zhao et al., 2022.

Liver failure

Liver failure was reported in 4 studies (Ishihara et al., 2022; Wang et al., 2014; Yang et al., 2024; Yu et al., 2020), and the combined effect size suggested that the liver failure rates in the underweight group was comparable to that in the normal weight group (RR 1.12, 95% CI [0.65–1.93], P = 0.68; heterogeneity: I2 = 0%) (Fig. 4A).

Bile leak

Bile leak was reported in six studies (Ishihara et al., 2022; Okamura et al., 2012; Wang et al., 2014; Yang et al., 2024; Yu et al., 2020; Zhao et al., 2022), and the combined effect size suggested that the bile leak rates in the underweight group was comparable to that in the normal weight group (RR 1.35, 95% CI [0.84–2.19], P = 0.22; heterogeneity: I2 = 0%) (Fig. 4B).

Surgical site infection

Surgical site infection (SSI) was evaluated in four studies (Ishihara et al., 2022; Okamura et al., 2012; Yang et al., 2024; Yu et al., 2020), and the pooled results showed that the SSI rates was higher in the underweight group than in the normal weight group (RR 2.00, 95% CI [1.03–3.88]; heterogeneity: I2 = 51%, P = 0.11) (Fig. 4C).

Blood transfusion

Blood transfusion was reported in four studies (Liu et al., 2022; Okamura et al., 2012; Wang et al., 2014; Yang et al., 2024), and the combined effect size suggested that underweight was not associated with an increase in transfusion rates (RR 1.14, 95% CI [0.74–1.77], P = 0.55; heterogeneity: I2 = 64%) (Fig. 4D).

Publication bias and sensitivity analysis

According to the funnel plots and Egger tests (Fig. 5), no evident publication bias was detected for the primary outcomes (mortality and overall morbidity). Sensitivity analysis showed that no single study affected the overall effect size of the mortality.

Figure 5 Funnel plot of underweight versus normal weight.

(A) mortality. (B) overall morbidity.

Trial sequential analysis

The cumulative Z-curves for SSI, total postoperative complications, operation time, and mortality distinctly surpassed both the trial sequential monitoring boundaries and the Required information size (RIS) boundaries (Fig. 6). Meanwhile, the Z-curve for blood loss, major complications, and length of the hospital stay (Fig. 7) only exceeded the RIS threshold, and the Z-curve for 90-day mortality, bile leak, liver failure, and blood transfusion (Fig. 8) failed to cross the RIS threshold. This suggests that conclusive evidence is available regarding the impact of underweight on SSI, total postoperative complications, operation time, mortality, blood loss, major complications, and the length of the hospital stay, while conclusive results for 90-day mortality, bile leak, liver failure, and blood transfusion remain elusive.

Figure 6 Trial sequential analysis of the association between underweight and (A) surgical site infection, (B) total postoperative complications, (C) operation time, and (D) mortality.

Figure 7 Trial sequential analysis of the association between underweight and (A) blood loss, (B) major complications, and (C) length of stay.

Figure 8 Trial sequential analysis of the association between underweight and (A) 90-day mortality, (B) bile leak, (C) liver failure, and (D) blood transfusion.

Discussion

Obesity has been reported to increase postoperative complications in patients with liver tumors and negatively affect their long-term prognosis (He et al., 2019; Shinkawa et al., 2018). However, few studies have evaluated the effect of preoperative underweight status on surgical outcomes in patients with liver tumors. Whether preoperative underweight in patients with liver tumors is associated with poorer postoperative short-term outcomes remains controversial. To our knowledge, this is the first meta-analysis to explore the effects of underweight on postoperative morbidity and mortality. Our results suggest that although there were no significant differences between the underweight and normal-weight groups in terms of intraoperative blood loss and length of hospital stay, underweight patients were at significantly higher risk of postoperative complications than normal-weight patients, especially SSIs. In addition, preoperative underweight was associated with increased postoperative mortality. Our study has important clinical value as we confirm, for the first time in a meta-analysis, that underweight status is a risk factor for morbidity and mortality after liver resection. This may help early identification of patients at high risk for postoperative complications.

Tumor growth can cause systemic inflammatory responses, which in turn lead to insulin resistance and hypercatabolism of protein and adipose tissue. These conditions can lead to significant nutrient consumption and weight loss (Zhao et al., 2018). Preoperative underweight has been shown to be an important independent risk factor for postoperative complications (Shida et al., 2021; Upala et al., 2016; Yin et al., 2024; Zhao et al., 2018). A meta-analysis of 12 studies included by Zhao et al. (2018) showed that gastric cancer patients with underweight had a significantly higher rate of postoperative complications than those with normal weight (RR: 1.28). Yin et al. (2024)’s retrospective study suggested that preoperative underweight was a risk factor for postoperative chylothorax of esophageal cancer. Similarly, the results of our meta-analysis suggest that preoperative underweight is associated with increased morbidity after liver resection. The higher incidence of postoperative complications in patients with preoperative underweight may be related to the following reasons. On the one hand, underweight patients are often accompanied by lymphocytopenia, suggesting suppression of immune function. On the other hand, patients with a low BMI are more likely to develop sarcopenia, which can impair their ability to adapt to stress and hunger during major surgery (Yang et al., 2024; Yu et al., 2020). In addition, underweight patients may be concomitant with cachexia, a well-known risk factor for postoperative complications (Chen et al., 2023).

SSI is the most common complication after liver resection, occurring in up to 25% of patients (Hede et al., 2015; Yang et al., 2024). Previous studies have shown that infectious complications are the most common non-cancer cause of postoperative death in patients (Zhao et al., 2018). SSI is associated with higher morbidity, longer hospital stays, and higher medical costs (Yang et al., 2024). In addition, postoperative SSI can increase the risk of tumor recurrence (Koike et al., 2024). Previous studies have suggested that patients who are underweight before surgery have a higher incidence of malnutrition, which delays the recovery of surgical incisions and increases the risk of SSI (Yin et al., 2024). Our pooled results also confirmed that preoperative underweight status significantly increased the risk of postoperative SSI (odds ratio (OR) 2.12). In addition, there were no significant differences between the underweight and normal-weight groups in terms of blood loss and length of hospital stay. Although the operation time was slightly prolonged in the preoperative underweight group, the 7.73 mins difference may not have an impact on clinical outcomes.

Several studies have suggested that preoperative underweight is associated with an increased risk of postoperative mortality (Upala et al., 2016; Zhao et al., 2018). A multicenter prospective cohort study of 118,707 patients undergoing non-bariatric general surgery showed that underweight was a risk factor for increased postoperative mortality (Mullen, Moorman & Davenport, 2009). Hede et al. (2015) found a significant increase in postoperative mortality in underweight colorectal cancer patients compared to normal-weight patients. This is similar to our findings, where we also found significantly higher postoperative mortality in the underweight group than in the normal-weight group. In addition, a retrospective study by Ishizuka et al. (2011) showed that preoperative underweight (BMI <22.5 kg/m2) was an important predictor of postoperative mortality.

Therefore, early identification of these underweight patients and intervention (such as nutritional support) may help improve short-term outcomes. A previous meta-analysis showed that preoperative oral or enteral immune modulating nutrition was effective in reducing postoperative infectious complications (OR 0.52) and shortening hospital stay (MD−1.57 days) in patients with gastrointestinal cancer (including liver cancer) (Adiamah et al., 2019). In addition, Yap et al. (2023)’s study showed that perioperative nutritional support could effectively increase the body weight of patients with liver cancer and reduce the incidence of postoperative SSI and ascites.

This study had several strengths. On the one hand, we have established strict inclusion criteria, defining underweight and normal weight according to the BMI classification standard of WHO, which not only ensures the homogeneity of the study, but also enhances the reliability of the results of this study. On the other hand, we conducted a comprehensive literature search, and there were no language or date restrictions, reducing potential bias.

Our study has the following limitations. First, most of the studies we included were retrospective, and the results of the analysis may be affected by the inherent limitations of retrospective studies. Several potential confounders (including cirrhosis, Child-Pugh or Model for End-Stage Liver Disease (MELD) scores, sarcopenia, cachexia, American Society of Anesthesiologists (ASA) status, and the complexity of liver resection) may have influenced the study outcomes. None of the included studies applied statistical methods (such as propensity score matching) to mitigate these confounders, and detailed data on these variables were often lacking. Consequently, adjusted estimates could not be extracted for meta-analysis, nor could sensitivity analyses based on these variables be performed. Cachexia and preoperative weight loss may mediate the observed effect, and the interplay between cirrhosis and low BMI could not be fully disentangled in this meta-analysis. Second, although our study showed that preoperative underweight was associated with increased postoperative morbidity and mortality, we were unable to assess the impact of preoperative underweight on long-term prognosis due to a lack of long-term follow-up studies. With few included studies, both trial sequential analysis and funnel plots should be interpreted cautiously. Our findings cannot be considered conclusive evidence, and publication bias cannot be definitively excluded. Furthermore, the lack of detailed reporting on surgeon experience and center volume further limits interpretation, as these factors are known to influence outcomes in complex liver surgery. Although BMI < 18.5 kg/m2 follows WHO standards, Asians may exhibit greater visceral adiposity at the same BMI compared with Western populations. Ideally, sarcopenia or skeletal muscle index would be evaluated, but such data were rarely available in the included studies. In addition, significant heterogeneity was observed in some of the outcomes. The diversity of populations and interventions may explain some of the high heterogeneity in the study outcomes. In our analysis, patients with Hepatocellular carcinoma (HCC) from Asian countries represented the majority of the sample. Only one study included patients with metastatic liver cancer, while the other six studies exclusively enrolled HCC patients. Therefore, our findings cannot be directly generalized to colorectal liver metastases or to Western populations. In addition, surgical approach, extent of resection, institutional volume, and geographic region are potential factors influencing the outcomes. However, due to the limited number of studies and lack of detailed stratified data, we were unable to conduct subgroup analyses to further explore these effects. Finally, some outcome measures are based on a combination of results from a small number of studies, which may limit the statistical power of this meta-analysis, and these results need to be treated with caution.

Conclusions

In conclusion, the results of this meta-analysis suggest that preoperative underweight is associated with an increased risk of postoperative complications, especially postoperative SSIs. In addition, patients who were underweight before surgery had a significantly higher mortality than those who were of normal weight. The absolute risk differences and number needed to treat for an additional harmful outcome (NNTH) highlight the clinical relevance of increased morbidity and mortality in underweight patients. Prospective studies or interventional trials, such as nutritional prehabilitation or stratification by cirrhosis and resection extent, are needed to confirm whether underweight represents a truly modifiable risk factor. Given the potential association between underweight status and adverse outcomes after hepatectomy, preoperative nutritional support should be considered for patients with underweight and poor nutritional status.

Supplemental Information

Supplemental Information 1 PRISMA 2020 abstract checklist

Supplemental Information 2 PRISMA checklist

We thank the authors of the included studies.

Additional Information and Declarations

Competing Interests

Author Contributions

Data Availability

The authors declare there are no competing interests.

Xuan Tang conceived and designed the experiments, performed the experiments, analyzed the data, prepared figures and/or tables, authored or reviewed drafts of the article, and approved the final draft.

Yunpeng Su conceived and designed the experiments, performed the experiments, analyzed the data, authored or reviewed drafts of the article, and approved the final draft.

Wenxi Li conceived and designed the experiments, analyzed the data, authored or reviewed drafts of the article, and approved the final draft.

Li Wan conceived and designed the experiments, performed the experiments, prepared figures and/or tables, authored or reviewed drafts of the article, and approved the final draft.

The following information was supplied regarding data availability:

This is a systematic review/meta-analysis.

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
