# Peer review of "Impact of preoperative underweight status on perioperative morbidity and mortality after liver resection for liver tumors: a meta-analysis"

_PeerJ, doi:10.7717/peerj.20324_

## Round 0.1 · original submission · Major Revisions

· Academic Editor

Major Revisions

Reviewers note some important methodological issues with your submission.

**Language Note:** The review process has identified that the English language must be improved. PeerJ can provide language editing services - please contact us at [email protected] for pricing (be sure to provide your manuscript number and title). Alternatively, you should make your own arrangements to improve the language quality and provide details in your response letter. – PeerJ Staff

·

Basic reporting

I read with interest the paper from Tang et Al. on the impact of underweight on liver resection outcome for liver tumors.
I believe the review work is well presented, clear, and within a well-defined scope. The presentation and rationale are sound, the language is comprehensible, and the results are in line with the literature in the field. The conclusion seems supported by the results, however, there are some methodological aspects that should be taken into account.

Experimental design

Search strategy
Especially for the PubMed search, no effort was made to expand the potential pool of results using MeSH terms in combination with the plain text terms. Since Pubmed search yeld just 30 results, I believe some more effort should have been done in identifying a potential larger body of evidence (for example, the simple adding of " "Thinness"[Mesh] OR "Body Mass Index"[Mesh]" yelds over 100 results, which are much more than the original 30, but still a manegeable amount).

Statistical analysis
The Authors state that "A random-effects model was employed if I² > 50%; otherwise, a fixed-effects model was used", referencing Higgins and Thompson (10.1002/sim.1186). This sentence is not supported by such paper. It is also a common methodological mistake. The model to use for analysis should be a priori defined, not depending on the data itself. This is because Random Effects and Fixed Effects are based on different assumptions. Random Effects outputs a distribution of "True" effects, sampled across different populations, while Fixed Effects estimates one "True" effect, coming from different resamplings of the same population. The second event almost never happens in meta-analysis in the field of Medicine. Cfr. Borenstein et al. (10.1002/jrsm.12), McKenzie et al. (10.1111/resp.12783). Worth mentioning, the PROSPERO protocol correctly only mentions the usage of the Random Effects model. Lastly, the results provide mean effect estimation, but no estimation of effect distribution; the prediction interval should also be provided. Cfr. IntHout et al. (10.1136/bmjopen-2015-010247).

3) Limitations
The Authors correctly cite that the low number of included studies may be a limitation for current findings, especially for secondary outcomes. However, in this case, Trial Sequential Analysis could help in defining whether it is a problem of information size or if we have already reached futility in the field. Cfr. Riberholt et Al. (10.1186/s12874-024-02318-y).

Validity of the findings

While the work has merits, current methodological flaws (potentially missed literature findings due to flawed search string, and questionable model choice for pooling of effect size) should be addressed before it is suitable for acceptance.

Reviewer 2 ·

Basic reporting

The language throughout the manuscript should be polished.

Abstract: What’s the significance of the comparison between patients in the obese category and those of normal weight?

Introduction: Please check the date of the complication rate following hepatectomy, which does not seem consistent with common sense.

Professional article structure, figures, tables.

Experimental design

The research question is well-defined, relevant & meaningful.

The mortality included in this study is not specified as 30-day or 90-day mortality. Although 30-day mortality has been validated, 90-day mortality is a more reliable outcome metric for measuring hospital performance and capturing procedure-related mortality.

Validity of the findings

The diagnosis of several studies is not limited to tumors, such as references 8, 11,12,15., which do not meet the inclusion criteria.

Additionally, the database included in the studies Robert 2009 and Mathur 2010 are the same as the American College of Surgeons National Surgical Quality Improvement Program (ACS NSQIP), and the periods also overlapped (2005-2006 & 2005-2008), which inevitably resulted in data with overlapping cohorts and redundant inclusion potentially.

---

## Round 0.2 · Major Revisions

· Academic Editor

Major Revisions

This manuscript addresses an important and clinically relevant question; however, several methodological aspects require clarification and strengthening before firm conclusions can be drawn. A first concern is causality and confounding. Most included studies are retrospective, and low BMI is strongly correlated with cirrhosis, Child-Pugh or MELD scores, sarcopenia, cachexia, ASA status, and the complexity of liver resection. Without a meta-analysis focusing preferentially on adjusted estimates, there is a significant risk that the observed associations are driven by these confounders rather than by low BMI itself. It is unclear whether the authors extracted and analyzed adjusted estimates when available or whether a sensitivity analysis was performed, restricting it to studies with adequate adjustments. Related to this, cachexia and preoperative weight loss may act as mediators of the observed effect, and the interplay between cirrhosis and low BMI is not disentangled in the current version.

Another issue concerns the heterogeneity of populations and interventions. Hepatocellular carcinoma patients from Asian countries heavily dominate the sample, and it is not clear whether the association holds for colorectal liver metastases or for Western populations. In addition, laparoscopic and open procedures, major and minor resections, and high- versus low-volume centers were combined. Subgroup analyses by etiology, geographic region, surgical approach, and extent of resection would help clarify the robustness and generalizability of the findings.

Statistical choices also deserve reconsideration. Odds ratios were used for relatively common outcomes such as postoperative complications, which can exaggerate the perceived effect size; risk ratios would be more clinically interpretable. With a small number of studies, the Hartung-Knapp-Sidik-Jonkman method could provide more conservative confidence intervals. For continuous outcomes such as operative time, the mean difference should be interpreted in terms of clinical significance—an increase of only a few minutes may not be meaningful in practice. Similarly, the trial sequential analysis and funnel plots have low reliability with so few studies; concluding that there is “no publication bias” or “conclusive evidence” is not justified under these circumstances unless the underlying assumptions (event rates, relative risk reduction, DARIS, alpha-spending) are explicitly presented and stress-tested.

Definitions of outcomes also raise questions. Were “major complications” consistently defined as Clavien-Dindo ≥III across studies? Were postoperative liver failure and bile leak defined according to ISGLS criteria? Was mortality consistently measured at 30 days, 90 days, or both? For surgical site infections, a discrepancy exists between the reported confidence intervals and the P-values, and clarification is needed on whether the reported P-values represent the effect or heterogeneity. Without harmonization of definitions, pooling results may be misleading.

Further, there is a potential risk of overlapping cohorts from multicenter studies or groups publishing multiple papers from the same patient pools. How did the authors ensure the independence of datasets and exclude duplicate patient populations? In addition, the Newcastle-Ottawa Scale assessment of study quality should be presented by domain and by study. A sensitivity analysis excluding studies with a high risk of bias in the comparability domain should also be reported.

The choice of BMI <18.5 kg/m² as a cutoff is in line with WHO standards; however, its applicability across populations with different body compositions remains uncertain. Asians may present higher visceral adiposity for the same BMI compared to Western populations. Did the authors explore alternative cutoffs or, where available, measures of sarcopenia such as skeletal muscle index? This would improve both the internal validity and the external generalizability of the findings.

Finally, the clinical significance of the observed associations should be emphasized. Beyond statistical significance, what are the absolute risk differences and numbers needed to harm for the key outcomes? Which results are clinically relevant (mortality, total morbidity) and which are trivial despite statistical significance (small increases in operative time)? A more cautious and nuanced conclusion is warranted, making clear the distinction between observational associations and causal inferences. The discussion should also address what prospective studies or interventional trials (for example, nutritional prehabilitation in underweight patients or stratification by cirrhosis and extent of resection) would be necessary to confirm whether low BMI is a truly modifiable risk factor in liver surgery.

---

## Round 0.3 · Minor Revisions

· Academic Editor

Minor Revisions

Thank you for submitting the revised version of the manuscript and for the work done so far. After a careful review of this new version, we have identified that, while progress has been made in many areas, there are still some points that can be improved to ensure that the work fully meets the quality and rigor standards expected by our journal.

We kindly ask that you consider these comments and revise the manuscript accordingly. We look forward to receiving the new version with these improvements.

Please do not hesitate to reach out if you have any questions or require further clarification.

·

Basic reporting

The Authors have greatly improved the manuscript following the reviewers' suggestions. I have no furhter remarks.

Experimental design

No comment.

Validity of the findings

No comment.

Additional comments

The Trial Sequential analysis figures may be misleading, since they are not scaled with the patients' number; therefore, understanding how far we are from the Required Information Size is not immediate. I suggest to change the scale of x axis to sample size instead of publication year, for better clarity of the image.

---

## Round 0.4 · accepted · Accept

· Academic Editor

Accept

Thank you for submitting the revised version of your manuscript. After carefully reviewing the changes, I can confirm that all reviewers' comments and suggestions have been appropriately addressed.

Based on this assessment, I am pleased to inform you that the manuscript is now ready for publication. Congratulations.